# Improved Sugarcane-Based Fermentation Processes by an Industrial Fuel-Ethanol Yeast Strain

**DOI:** 10.3390/jof9080803

**Published:** 2023-07-29

**Authors:** Gabriela Muller, Victor R. de Godoy, Marcelo G. Dário, Eduarda H. Duval, Sergio L. Alves-Jr, Augusto Bücker, Carlos A. Rosa, Barbara Dunn, Gavin Sherlock, Boris U. Stambuk

**Affiliations:** 1Department of Biochemistry, Federal University of Santa Catarina, Florianópolis, Santa Catarina 88040-900, Brazil; muller.gabriela@gmail.com (G.M.); victor@picway.com.br (V.R.d.G.); marcelogdario@hotmail.com (M.G.D.); eduardahd@hotmail.com (E.H.D.); slalvesjr@gmail.com (S.L.A.-J.); abucker@gmail.com (A.B.); 2Departamento de Microbiologia, Universidade Federal de Minas Gerais, Belo Horizonte, Minas Gerais 31270-901, Brazil; carlrosa@icb.ufmg.br; 3Department of Genetics, Stanford University, Stanford, CA 94305, USA; bdunn@stanford.edu (B.D.); gsherloc@stanford.edu (G.S.)

**Keywords:** bioethanol, sugarcane, yeast, fermentation

## Abstract

In Brazil, sucrose-rich broths (cane juice and/or molasses) are used to produce billions of liters of both fuel ethanol and *cachaça* per year using selected *Saccharomyces cerevisiae* industrial strains. Considering the important role of feedstock (sugar) prices in the overall process economics, to improve sucrose fermentation the genetic characteristics of a group of eight fuel-ethanol and five *cachaça* industrial yeasts that tend to dominate the fermentors during the production season were determined by array comparative genomic hybridization. The widespread presence of genes encoding invertase at multiple telomeres has been shown to be a common feature of both baker’s and distillers’ yeast strains, and is postulated to be an adaptation to sucrose-rich broths. Our results show that only two strains (one fuel-ethanol and one *cachaça* yeast) have amplification of genes encoding invertase, with high specific activity. The other industrial yeast strains had a single locus (*SUC2*) in their genome, with different patterns of invertase activity. These results indicate that invertase activity probably does not limit sucrose fermentation during fuel-ethanol and *cachaça* production by these industrial strains. Using this knowledge, we changed the mode of sucrose metabolism of an industrial strain by avoiding extracellular invertase activity, overexpressing the intracellular invertase, and increasing its transport through the *AGT1* permease. This approach allowed the direct consumption of the disaccharide by the cells, without releasing glucose or fructose into the medium, and a 11% higher ethanol production from sucrose by the modified industrial yeast, when compared to its parental strain.

## 1. Introduction

Sugarcane is a predominant crop in tropical countries and is used as raw material for several fermentation processes employing *Saccharomyces cerevisiae* yeasts. Bioethanol, mainly used as an automotive fuel, is by far the most common renewable fuel, and it is thus poised to contribute greatly to goals of energy independence and environmental sustainability [1]. Fuel-ethanol production from sugarcane represents the major large-scale technology capable of producing bioethanol efficiently and economically. Indeed, Brazil is one of the most competitive producers of bioethanol in the world, having the lowest cost of production worldwide and accounting for approximately ~30% of the global production, with almost 27 billion liters produced in 2022 from sugarcane [2,3,4]. Sugarcane juice and molasses can also be used for the production of baker’s yeast and several distilled alcoholic beverages. For example, in Brazil, the sugarcane juice is used to produce *cachaça*, a distilled spirit that nowadays is the third most popular distilled beverage in the world, with an annual production reaching almost 2 billion liters, performed by industrial and thousands of artisanal producers [5].

The Brazilian industrial process of bioethanol production is characterized by fermentations in very large tanks (0.5 to 3 million liters) using very high yeast-cell densities (10–15% *w*/*v*) in a fed-batch mode (the so-called Melle–Boinot process) to ferment sugarcane juice and/or diluted molasses, which contain up to 150–200 g/L of total sugar [6,7]. The yeast cells are collected by centrifugation after each fermentation cycle of 8–10 h, treated with diluted sulfuric acid for 1–2 h, and more than 90% of the yeasts are reused from one cycle to the next. This ensures the high cell density that contributes to very short fermentation times and thus high productivity of the process. *Cachaça* production also involves the fermentation of sugarcane juice containing 140–160 g/L of total sugar with a starter culture from a previous fermentation, which normally represents 20% of the total volume of the fermentation vat [8]. Both bioethanol and *cachaça* production are non-sterile processes, and thus subject not only to contamination by microorganisms, but also to a continuous succession of yeast strains [9,10,11,12], some of which can dominate the fermenters and contribute to efficient and stable fermentations [12,13,14].

The disaccharide sucrose (α-d-glucopyranosyl-(1→2)-β-d-fructofuranoside) is by far one of the cheapest sugars in the industrial utilization of the yeast *S. cerevisiae* [15]. It is well known that yeast cells harbor an extracellular invertase (β-d-fructosidase) that hydrolyzes sucrose into glucose and fructose, which are transported by facilitated diffusion into the cells by hexose transporters and metabolized through glycolysis. Invertase is encoded by one or several telomeric *SUC* genes (*SUC1* to *SUC5* and *SUC7* to *SUC10*). *SUC2* is the most common locus found in the sub-telomeric region of chromosome IX of almost all *S. cerevisiae* strains, including other closely related yeast species, and thus it is considered the ancestral locus of this gene family [16,17]. Since the amplification of telomeric *SUC* genes is only observed in industrial (e.g., bakers’, brewers’, and distillers’ yeasts) strains, it was suggested that this trait was under selection during the process of domestication of these yeasts [18,19]. These *SUC* genes generate two different mRNAs: a larger transcript encoding an invertase with a signal sequence required for its secretion from the cell, and a shorter transcript lacking this signal sequence and thus coding for an intracellular form of the enzyme [20]. While the former mRNA is repressed by high concentrations of sucrose or its hydrolysis products (glucose and fructose), the intracellular invertase is expressed constitutively. Finally, efficient extracellular invertase expression requires low levels of glucose or fructose in the medium [21]. The transcriptional activator of this gene is still unknown despite significant improvements in our knowledge regarding the molecular mechanisms involved in the repression of *SUC* expression [22].

Although high invertase activities would be expected to ensure efficient fermentation of sucrose, several reports have shown poor (or even negative) correlations between the levels of this enzyme at the yeast surface and the fermentative performance of the cells [23,24]. Indeed, it has been shown [25,26] that the vast majority (>90%) of the glucose and fructose molecules resulting from sucrose hydrolysis diffuse away into the medium before they can be imported into the yeast cell, which may allow the growth of other contaminant microorganisms, including those lacking invertase or that are unable to ferment this disaccharide. However, it is important to note that sucrose is both a β-fructoside and a α-glucoside, and thus it can also be metabolized through the active transporters and enzymes responsible for fermentation of maltose (and other α-glucosides) in yeasts [27,28,29,30,31]. This alternative pathway is interesting from a biotechnological point of view due to the energy required for the H^+^-sucrose symport activity, which consumes one ATP molecule per sucrose transported. This results in a lower biomass, which can be compensated for by enhanced consumption of the sugar, leading to higher ethanol yields when compared with strains hydrolyzing sucrose outside the cells [32,33,34].

Several other strategies to reduce the ATP yield during sugar fermentation have been developed more recently, including, for example, the overexpression of bacterial genes of the ATP-hydrolyzing F_1_-part of the ATPase enzyme [35], overexpression of an alkaline phosphatase [36], or even expression of hexose-H^+^ symporters from other yeasts [37]. All these studies reported lower biomass biosynthesis and increased (10–17%) ethanol production by the modified yeast strains, which, in many cases, required extensive laboratory evolution approaches in anaerobic conditions in order to obtain strains with the desired characteristics [32,35,37]. All these studies were performed with laboratory *S. cerevisiae* strains that, while being suited to be used in “proof-of-concept” approaches, probably have little (or no) industrial application. Only the report by Semkiv et al. [36] also presented a strategy implemented in an industrial strain commercially used in bioethanol production, and when this strain was modified to overexpress an alkaline phosphatase, it produced only 6% more ethanol from 20% glucose—less than the values obtained with laboratory strains used by the same authors. Indeed, recent work shows that strategies developed with laboratory strains (e.g., increasing the tolerance to ethanol) do not always work equally in industrial yeast strains [38,39].

Thus, the phenotypic and genetic characteristics of a group of eight fuel-ethanol and five *cachaça* industrial yeasts were analyzed to develop a strategy to genetically modify an industrial fuel-ethanol strain for improved ethanol yield from sucrose. Our array comparative genomic hybridization (array-CGH) results show that only two strains (one fuel-ethanol and one *cachaça* yeast) have amplification of the genes encoding invertase, reflected by a high invertase activity. The other industrial yeast strains had a single locus (*SUC2*) encoding invertase in their genome, although with different patterns of invertase activity. Based on these results, one strain was chosen, and an integrative genomic modification approach was developed to avoid extracellular invertase activity, overexpress the intracellular form of invertase, and improve the activity of the *AGT1* permease responsible for high-affinity H^+^-sucrose symport activity [27,29]. The modified industrial yeast strain consumed sucrose directly, without releasing glucose or fructose into the medium, and produced more ethanol when compared to the parental unmodified yeast strain.

## 2. Materials and Methods

### 2.1. Strains, Media, and Growth Conditions

The *S. cerevisiae* strains analyzed in this study are listed in Table 1. Yeasts were grown on rich YP medium containing 20 g/L peptone and 10 g/L yeast extract supplemented with 20 g/L glucose, 20 g/L sucrose, or 20 g/L maltotriose. When required, 20 g/L agar, 0.5 g/L zeocin (Invivogen, San Diego, CA, USA), 3 mg/L Antimycin A, or 200 mg/L geneticin (G-418) sulfate (both from Invitrogen, Thermo Fisher Scientific Inc., Sinapse Biotecnologia, São Paulo, SP, Brazil) were added to the medium. The pH of the medium was adjusted to pH 5.0 with 1 M HCl, except when using zeocin, where the pH was adjusted to pH 8.0 with 0.5 M NaOH. Cells were grown aerobically at 28 °C with shaking speed of 160 rpm in cotton-plugged Erlenmeyer flasks filled to 20% of the volume with medium, and cellular growth was followed by turbidity measurements at 570 nm (OD_570nm_).

### 2.2. Array-CGH Protocol

The array comparative genome hybridization analysis of the industrial yeast strains was performed as described previously [41,44]. Microarrays onto which had been spotted PCR products corresponding to full-length ORFs from the laboratory S288C strain [45] were used, and thus the reference DNA in all hybridizations was isolated from this yeast. Genomic DNA was isolated with YeaStar columns (Zymo Research, Irvine, CA, USA), cut with *Hae*III (New England Biolabs, Ipswich, MA, USA), and 1 mg of this DNA was labeled with fluorescently tagged nucleotides (Perkin-Elmer, Waltham, MA, USA), usually Cy3-dUTP for the reference strain and Cy5-dUTP for the industrial strains, using the BioPrime random-prime labeling system (Invitrogen). After labeling, the reactions were heat-inactivated, the experimental and reference DNAs were mixed, purified away from unincorporated label using Zymo Clean&Concentrate columns (Zymo Research), and then hybridized to the microarrays at 65 °C as described [41,44]. Arrays were scanned with an Axon 4000 A scanner and the data were extracted using GenePix (Molecular Devices Corp., Union City, CA, USA) software. The array-CGH data were treated and analyzed as described previously [41,44]. Appendix A lists all genetic elements present in the microarrays and their significant (or not) red/green (R/G) ratios, obtained with the CGH-Miner program [46].

### 2.3. PFGE, Chromosome Blotting and Hybridization

Yeast chromosomes were prepared from 1 mL of yeast cells pre-grown in YP-20 g/L glucose medium and collected at the stationary phase of growth. Cells were treated with Zymolyase 20T and proteinase K in low-melting-point agarose blocks, transferred to a 1% agarose gel, and pulsed-field gel electrophoresis (PFGE) was performed at 8 °C using a Gene Navigator system (Amersham Pharmacia Biotech do Brasil Ltd.a., São Paulo, SP, Brazil) as previously described [29,40,41,44]. Following electrophoresis, the gel was stained with ethidium bromide and photographed (Gel Doc™ XR, BioRad Laboratories, Hercules, CA, USA). The chromosomes separated by PFGE were transferred to a nylon membrane (Hybond-N^+^, GE Healthcare, Barueri, SP, Brazil) by capillary blotting. Labeling of DNA probes (see below), pre-hybridization, hybridization, stringency washes, and chemiluminescent signal generation and detection were performed with an AlkPhos kit (GE Healthcare) as recommended by the manufacturer. After hybridization, an autoradiography film (Hiperfilm™ ECL, Kodak, GE Healthcare) was exposed to the membrane for 1 to 2 h before it was developed. Images were obtained with Image Lab Software (Gel Doc™ XR) and annotated with Microsoft PowerPoint. Probes were generated by PCR using DNA from strain S288C as template with primers (Table 2) SUC100-F and SUC1320-R for the *SUC2* gene.

### 2.4. Determination of Invertase Activity

Yeast cells were incubated for 12 h in rich YP medium containing 20 g/L sucrose, 20 g/L ethanol plus 30 g/L glycerol (derepressed conditions), or in derepressed conditions with media supplemented with 1 g/L glucose, and centrifuged (3000× *g*, 2 min). The extracellular (periplasmic) invertase activity was determined with whole cells pre-incubated with 50 mM NaF to block glycolysis as described [49], using 100 mM of sucrose. The total invertase activity (extracellular plus intracellular) was determined with permeabilized yeast cells [50] and also 100 mM of sucrose as substrate. The intracellular invertase activity was calculated by subtracting the periplasmic invertase from the total invertase activity. The invertase activity was expressed as nmol of glucose produced (mg dry cell weight-DCW)^−1^ min^−1^.

### 2.5. Determination of the Activity of the AGT1 Permease

The activity of the *AGT1* permease was determined with a colorimetric assay using *p*-nitrophenyl-α-D-glucopyranoside (*p*NPαG) as substrate [51]. The *p*NPαG transport activity was expressed as nmol of *p*-nitrophenol produced (mg DCW)^−1^ min^−1^.

### 2.6. Molecular Biology Techniques

Standard methods for bacterial transformation, DNA manipulation, and analysis were employed [52]. To overexpress the intracellular form of invertase in *S. cerevisiae*, a laboratory strain was transformed with the *kanMX*-P_ADH1_ module from plasmid pFA6a-*kanMX*6-P_ADH1_ [48], which introduced the strong constitutive *ADH1* promoter in front of the second codon encoding methionine of the *SUC2* gene, after 60 bp of the first start codon, removing the 20 N-terminal amino acids of the resulting protein [32]. The DNA of strain BSY21-34B3 (Table 1) overexpressing the intracellular form of invertase was used to amplify a DNA fragment containing long regions of homology (635 bp and 459 bp, respectively) to the upstream and downstream region of the *kanMX*6-P_ADH1_::*iSUC2* module using primers V2-SUC2F and V2-SUC2R. After transforming [53] cells of the industrial strain CAT-1 with this module, the cells were plated on YP-20 g/L glucose medium containing G-418 and incubated at 28 °C. The G-418-resistant isolates were tested for proper genomic integration of the *kanMX*6-P_ADH1_::*iSUC2* cassette at the *SUC2* locus by diagnostic colony PCR using primers V3-SUC2F and SUC100-R (Table 2), which amplified a 3274 bp fragment from the *kanMX*6-P_ADH1_::*iSUC2* locus, and a 774 bp fragment from a normal *SUC2* gene (since CAT-1 is diploid, both fragments were obtained). Next, part of the normal copy of the remaining *SUC2* gene was deleted using primers ssSUC2-F1 and 551SUC2-R1, which allowed the amplification of the *Ble*^R^ gene from plasmid pUG66 (Table 2) having 60 bp of homology to the beginning of the *SUC2* gene (absent in the *kanMX*6-P_ADH1_::*iSUC2* locus) and to the position 551 in the coding region of the *SUC2* gene, thus truncating almost 1/3 of the normal *SUC2* open-reading frame. The resulting PCR product was used to transform competent yeast cells, and zeocin-resistant isolates were tested for proper genomic integration of the *Ble*^R^ gene at the *suc2*Δ locus by analytical colony PCR using primers V-SUC2F and SUC100R (Table 2), which amplified a ~2700 bp fragment from the *kanMX*6-P_ADH1_::*iSUC2* locus, but failed to amplify a 550 bp fragment that would be obtained if the strain had a normal *SUC2* gene (e.g., strain CAT-1).

Finally, to overexpress the high-affinity sucrose-H^+^ symporter encoded by the *AGT1* gene [27,28,29], primers GPD-AGT1F and AGT1-389R (Table 2) were used to amplify from plasmid pGRSd-AGT1 (Table 2 [47]) the P_GPD_ (P_TDH3_) promoter region controlling the *AGT1* gene present in the plasmid. The linear DNA fragment contained an upstream sequence with 55 bp of homology to the upstream region of the *AGT1* gene present in chromosome VII (~279 bp from the start of the gene), the P_TDH3_ promoter, and 409 bp of the beginning of the *AGT1* ORF (and thus acting as a long region of homology for integration). The isolation of transformants was performed using YP-20 g/L maltotriose plates containing 3 mg/L Antimycin A as selecting media, since in the presence of Antimycin A, strains that do not have a functional *AGT1* permease would not grow (e.g., strain CAT-1) because they grow aerobically on this carbon source [44]. The correct integration of the P_TDH3_ promoter upstream of the *AGT1* gene was verified in the selected strains using primers V-GPDF and AGT1-I505R (Table 2), which allowed amplification of a 1944 bp fragment from strain GMY08 (but not from strain CAT-1), confirming the correct modification of the promoter region of the *AGT1* gene. Since the strains are diploids, two other primers (V-AGT1F1 and AGT1-I505R, Table 2) were used to check if the two copies of the *AGT1* were modified. The region with homology to primer V-AGT1F1 was removed with the P_TDH3_-AGT1 module, but since strain GMY08 still had a 1583 bp fragment amplified with these two primers, it indicated that only one copy of the *AGT1* gene was modified (as expected, strain CAT-1 also had this fragment amplified when analyzed by colony PCR).

### 2.7. Sucrose Batch Fermentations

Yeast strains were grown in rich YP medium containing 20 g/L sucrose, collected at the end of the log phase (<1 g DCW/L), centrifuged (3000× *g*, 5 min), washed twice with ice-cold distilled water, and resuspended in ice-cold distilled water in order to obtain a suspension with 20 g DCW/L. A volume of this suspension was mixed with the same volume of 2x YP medium (40 g/L peptone and 20 g/L yeast extract) containing 400 g/L sucrose in order to simulate industrial conditions: high cell densities (10 g DCW/L) and high sucrose concentrations (~200 g/L). The fermentations were carried out in cotton-plugged Erlenmeyer flasks filled to 1/5 of the volume, with shaking speed of 160 rpm and at 28 °C. One mL culture samples were harvested regularly, the OD_570nm_ determined, centrifuged (5000× *g*, 2 min), and their supernatants stored at −20 °C for the determination of sugars and ethanol. The chemical and biochemical methods used to quantify the concentrations of sucrose, glucose, fructose, and ethanol have been described in detail before [54]. For the comparison between strains CAT-1 and GMY08, the concentrations of sucrose, glucose, fructose, ethanol, and glycerol were determined by high-performance liquid chromatography (HPLC) equipped with a refractive index detector (RI-2031 Plus; JASCO, Tokyo, Japan) using an HyperREZ XP Organic Acid Column (Thermo Scientific, Waltham, MA, USA). The HPLC apparatus was operated at 20 °C using 5 mM H_2_SO_4_ as the mobile phase at a flow rate of 0.2 mL/min.

## 3. Results and Discussion

### 3.1. Microarray Karyotyping of Industrial Yeast Strains Used in Sugarcane-Based Fermentation Processes

The aCGH analysis was performed with thirteen industrial yeast strains that are used in the sugarcane-based fermentation process in Brazil (Table 1): eight industrial fuel-ethanol strains—six from the State of São Paulo (strains BG-1, CAT-1, PE-2, SA-1, and VR-1, previously analyzed by Stambuk and co-workers [41]) plus strain BAT, and two yeast strains isolated from the State of Paraíba (strains UFPE-135 and UFPE-179); and five *cachaça* yeast strains isolated from the State of Minas Gerais (strains UFMG-829, UFMG-905, UFMG-1007, UFMG-2097, and UFMG-2439). Although strain BAT showed some undesired industrial characteristics, such as excessive foam production, flocculation, and premature sedimentation [40], it is a yeast strain that dominates the industrial process. Strain UFMG-905, besides being used as a starter in *cachaça* production [13,42], is also a yeast strain with interesting probiotic characteristics [55,56].

Visualization of the data with the Java TreeView program [57] revealed that all yeast strains analyzed showed hybridization patterns with signals across most of the chromosomes corresponding to R/G ratios near a value of 1.0, indicating that the genomes of these industrial diploid yeast strains lacked chromosomal aneuploidies (see also Appendix A). Figure 1A shows the array-CGH data for a selection of genes involved in sugar fermentation, including the *SUC2* gene encoding invertase and 20 genes involved in sugar transport (*HXT1* to *HXT17*, plus *GAL2* and the transceptors *SNF3* and *RGT2*). In addition, Figure 1B shows the data for genes involved in thiamine and pyridoxine biosynthesis (*SNO*/*SNZ* genes), which were previously shown to be amplified in the five fuel-ethanol industrial strains from São Paulo [41], as well as genes involved in the transport of thiamine and its precursors (*THI7*, *NRT1*, and *THI72* genes).

From the data shown in Figure 1A, it is evident that, with the exception of the fuel-ethanol strain UFPE-135 and the *cachaça* strain UFMG-829, which showed amplification of the *SUC2* gene, all the strains seem to have the usual copy number of this gene in their genome. Almost all strains also have regular copy numbers of the major hexose transporters *HXT1* to *HXT7* [58,59,60]. On the other hand, a lower copy number was detected for the telomeric *HXT* genes *HXT8*, *HXT9*, *HXT11*, *HXT12*, and *HXT14* to *HXT17* for almost all of the industrial strains analyzed. Note that *HXT10* is not telomeric, while *HXT13* is located at one of the telomeres of chromosome V. The genes *HXT13*, *HXT15*, *HXT16*, and *HXT17* lost the status of hexose transporters since they were reported as polyol (xylitol, mannitol and sorbitol) transporters [61]. Although all strains seem to have a lower copy number of the galactose-inducible *GAL2* permease; growth on this carbon source by many of the strains was not impaired by the presence of the mitochondrial inhibitor antimycin A, and thus the strains still have a functional *GAL2* transporter in their genomes [62]. Many of the telomeric *HXT* transporters absent in the genome of the industrial strains are probably the consequence of their replacement by the amplified *SNO*/*SNZ* genes (also telomeric) involved in pyridoxine and thiamin biosynthesis (Figure 1B) [41,63]. In strain S288C, these genes are located in the left telomere of chromosome VI (*SNO3*/*SNZ3*) and chromosome XIV (*SNO2*/*SNZ2*). The industrial sugarcane strains, which lack the genes in chromosome VI, have these genes spread in the telomeres of chromosomes IV, VII, IX, X, XI, and/or XIV [41,64]. The amplified *SNO*/*SNZ* genes present in chromosome IV replace the telomeric region containing the *HXT15* gene. In the case of chromosome X, there is the particular situation in some industrial strains in which both telomeres might contain the *SNO*/*SNZ* genes, in the right telomere (replacing the region containing the *HXT16* gene) and also in the left telomere (replacing the region containing the *HXT8* and/or *HXT9* genes) [64,65]. Figure 1 also shows that, in some industrial strains (e.g., the *cachaça* yeasts), the *SNO*/*SNZ* genes (particularly the *SNZ2*/*SNZ3* genes) were not amplified as much as in the case of the fuel-ethanol strains. Interestingly, these strains do show amplification of transporters involved in the uptake of thiamin (*THI7* gene) and its precursors 2-methyl-4-amino-5-hydroxymethylpyrimidine (derived from vitamin B6) and/or 4-methyl-5-β-hydroxyethylthiazole, and the *NRT1* (also known as *THI71*) and *THI72* genes [66,67]. These results highlight the importance that thiamin (and also pyridoxine) has for yeast strains used in sugarcane-based fermentation processes, as well as other industrial fermentations (e.g., production of wine) that normally use high sugar concentrations [41,68].

Analysis of genomic variation among a large number of *S. cerevisiae* strains has revealed not only significant genetic variability but has also shown that, in general, these strains cluster according to their technological application rather than geographical origin [69,70,71], indicating that wild and domesticated populations have evolved different life strategies for adaptation to generally different environments. Although there are different driving forces that shape the genome structure and gene content of domesticated and wild yeasts, variations in gene copy number have a greater phenotypic effect than do single nucleotide polymorphisms or other genomic changes [71,72]. In the case of the Brazilian *cachaça* yeast strains, their genomic signatures indicate that they derive from wine yeasts that have undergone an additional round of domestication, or “secondary domestication” [73], while the industrial fuel-ethanol yeast strains are also closely related to *cachaça* strains forming a single monophyletic clade proximal to wine strains [64,71,74]. Our array-CGH results indeed confirm the close proximity of the *cachaça* and fuel-ethanol strains, sharing, for example, common loss of some telomeric *HXT* genes, while genes involved in thiamine (B1) and pyridoxine (B6) biosynthesis and uptake from the media are amplified in all these sugarcane fermenting yeasts.

### 3.2. Invertase Activity of Industrial Yeast Strains Used in Sugarcane-Based Fermentation Processes

The aCGH analysis of the thirteen industrial yeast strains indicated that only two strains (strain UFPE-135 and UFMG-829) showed amplification of the *SUC2* gene encoding invertase. Figure 2 shows a chromosomal blot of the *SUC* loci found in the laboratory S288C strain (*SUC2*), as well as in some of the fuel-ethanol yeast strains, including strain UFPE-135, which has a higher copy-number of *SUC* genes in its genome. It is evident that the majority of sugarcane yeast strains harbors only the *SUC2* gene in their genomes (strains CAT-1, PE-2, BG-1, and SA-1), while strain UFPE-135 has, besides this gene, the *SUC1* gene located in chromosome VII. The identity of the VII chromosome copy was confirmed by hybridization with a probe based on the *AGT1* gene (see [44]). Other strains (e.g., UFMG-1007, UFPE-179, and VR-1) also presented only the *SUC2* loci in their genomes (see Appendix A).

When the invertase activity was determined in these yeast strains, many of them (the fuel-ethanol strains CAT-1, PE-2, VR-1, and UFPE-179, and the *cachaça* yeast strain UFMG-1007) had a similar profile to that of the laboratory strain S288C: higher activity on sucrose compared to the derepressed conditions (growth with ethanol/glycerol), and even higher activity after growth on the derepressed conditions supplemented with 0.1% glucose, one of the best conditions for induction of invertase expression [75]. Some strains showed a slightly different pattern, with a high invertase activity after derepression conditions (strains BG-1 and SA-1), and as expected, the strains that had amplification of the *SUC* genes (strains UFMG-829 and UFPE-135) were the ones with the highest invertase activity under all conditions tested (Figure 3).

### 3.3. Sucrose Batch Fermentation by the Industrial Yeast Strains Used in Sugarcane-Based Fermentation Processes

Sucrose batch fermentations trying to simulate industrial conditions were performed with initial high (>200 g/L) sucrose concentrations to be fermented by high (10 g/L DCW) cellular concentrations. Figure 4 shows the results for some strains (PE-2, BG-1, UFPE-179, and UFMG-1007) that have a typical pattern of sucrose consumption: sucrose is hydrolyzed rapidly and totally consumed in ~6 h (strains PE-2, UFPE-179, and UFMG-1007), releasing significant amounts of glucose and fructose into the medium, which will be consumed by ~8 h of fermentation to produce 68–78 g/L ethanol. Note that due to higher invertase activity, strain BG-1 hydrolyzes almost all sucrose in ~4 h, releasing higher amounts of glucose and especially fructose into the medium. In general, fructose accumulates to higher concentrations and takes longer to be consumed by the yeast cells, when compared to the glucose released into the medium (Figure 4). In the case of the yeast strains UFMG-829 and UFPE-135 with amplification of the *SUC* genes (Figure 1), with high invertase activity (Figure 3), the hydrolysis of sucrose is so fast that during the initial mixing and sampling of the fermentation broth (which included a 2 min centrifugation) a significant release of glucose, fructose, and ethanol occurs, and sucrose is totally hydrolyzed in less than 4 h (Figure 5), producing 74–77 g/L of ethanol.

### 3.4. Modifying the Mode of Sucrose Fermentation by an Industrial Fuel-Ethanol Yeast Strain

The results shown in the previous sections indicate that all the industrial strains analyzed contain common genomic features, many of them probably contributing to their dominance in the sugarcane-based industrial processes. Regarding sucrose fermentation, the majority of strains have a single invertase locus (*SUC2* gene) in their genome, which is an important characteristic if the idea is to modify the way that the yeast cells will ferment sucrose: instead of its extracellular hydrolysis, the disaccharide will be actively transported into the cell and hydrolyzed intracellularly [32]. To achieve these goals, initially it was necessary to modify the genes encoding invertase; given that the industrial strains are diploid, this requires modifying two copies, even in those strains having a single *SUC2* locus. Thus, the genetic engineering strategies were performed with strain CAT-1, which has previously been very well characterized in many aspects, including its genome sequence [76].

First, one copy of the *SUC2* gene had its promoter region modified in order to overexpress the intracellular form of invertase (*kanMX*6-P_ADH1_::*iSUC2*), placing the constitutive P_ADH1_ promoter controlling the expression of an invertase lacking the 20 first amino acids, the signal peptide that drives the protein to the endoplasmic reticulum and Golgi for its secretion [20]. The other copy of the *SUC2* gene was partially deleted (*suc2*Δ::*Ble*^R^), thus avoiding the expression of the extracellular form of invertase. Table 3 shows that strain CAT-1 had an extracellular invertase activity ~4 times higher than the intracellular invertase activity. On the other hand, the opposite trend was observed for the modified strain GMY08, with a large increase in intracellular invertase activity and a significant decrease in the extracellular invertase activity. Note that the residual “extracellular invertase” activity of strain GMY08 is probably the consequence of sucrose entrance into the cells, and its intracellular hydrolysis, releasing glucose molecules that cannot be metabolized by the yeast cells due to blockage of glycolysis by sodium fluoride [47].

It was evident from our previous work on engineering the mode of sucrose fermentation in laboratory yeast strains that sucrose transport by the *AGT1* permease was a limiting factor [32]. Only after a laboratory evolution approach in anaerobic sucrose-limited chemostats was it possible to isolate an evolved strain with higher transport activity, due to a duplication of the *AGT1* gene, which showed the expected improved sucrose fermentation performance [32]. Strain CAT-1 has the *AGT1* gene in chromosome VII [76], but this strain does not efficiently ferment maltotriose (another substrate of the *AGT1* permease) [44] due to the presence of a divergent and non-functional promoter region, also found in other industrial yeast strains unable to ferment maltotriose [77]. Instead, strain CAT-1 uses maltotriose only after an extensive lag phase, and it metabolizes this sugar aerobically due to its extracellular hydrolysis mediated by the isomaltase encoded by the *IMA5* gene [44,78]. Indeed, as can be seen in Table 3, strain CAT-1 had no *p*NPαG transport activity, a measurement of this permease’s activity, indicating that it was also necessary to overexpress the *AGT1* gene. For this purpose, the promoter region of the *AGT1* gene present in CAT-1 was replaced by the constitutive P_TDH3 _promoter using the same strategy of Vidgren and co-workers [79], and the transformants were selected in maltotriose plates containing Antimycin A. Although only one copy of the *AGT1* gene had its promoter modified in strain GMY08 (P_TDH3_::*AGT1*), it clearly had an increased activity of this transporter (Table 3). It is important to note that overexpression of the *AGT1* permease is certainly a required modification in the fuel-ethanol yeasts, since this gene is normally regulated by maltose or maltotriose through the *MAL* regulatory system [80], and not by the presence of sucrose. Only a strain that is *MAL* constitutive, as some laboratory strains used previously (strains CEN.PK2-1C or 1403-7A, see [29,32]), will allow the expression of the *AGT1* permease during growth on sucrose.

The physiological performance of the GMY08 strain was evaluated in sucrose batch fermentation and compared to the parental strain (Figure 6). Strain CAT-1 showed a fermentation performance similar to that of other industrial sugarcane strains (see Figure 4), hydrolyzing sucrose in up to 5–6 h and releasing huge amounts of hexoses that took up to 8 h to be completely consumed. The modified strain GMY08 showed a slower sucrose consumption for 6–7 h, without releasing hexoses in the medium (Figure 6). Because of the energy demand of the active sucrose-H^+^ symport [32], it produced approximately ~11% more ethanol than the parental CAT-1 strain. In addition, the amount of glycerol produced by strain GMY08 during sucrose batch fermentation was half of that produced by the parental strain CAT-1 (Table 3). Besides the expected higher ethanol production as a consequence of the energetics of sucrose transport [32], strain GMY08 did not release in the medium any glucose or fructose, sugars that can favor microbial contamination even by microorganisms that would not normally use sucrose as a carbon source [25,26,81]. Another consequence of the hydrolysis of high sucrose concentrations by invertase is a sudden osmotic shock promoted by the high glucose plus fructose concentration in the medium. Yeast cells respond to the osmotic stress by increasing the production of glycerol from the glycolytic intermediate dihydroxyacetone phosphate, which will lead to a decrease in the ethanol yield [24,82,83,84,85].

## 4. Practical Implications of this Study

In order to improve sugarcane-based processes, it is necessary to identify the genomic and physiological adaptations that the industrial yeast strains have in order to survive in the harsh industrial process. We found that the group of 13 industrial strains studied shared common genomic features, and 11 of them had only the *SUC2* loci in their genome, allowing the development of a genomic engineering strategy to modify the mode of sucrose fermentation by yeast. While the engineered industrial strain developed in this study showed good fermentation performance on sucrose, when compared to the wild-type strain, it will be important to evaluate its fermentation performance under industrial process conditions, including its possible effect on the microbial dynamics in the fermentation tanks, the effects of up-scaling into huge fermentation tanks, and how this modified strain will behave during centrifugation and recycling of the cells, a practice that takes place during the whole crop season to ensure the high cell densities that contribute to the short fermentation times currently in practice [6,7,74]. 

## Figures and Tables

**Figure 1 jof-09-00803-f001:**
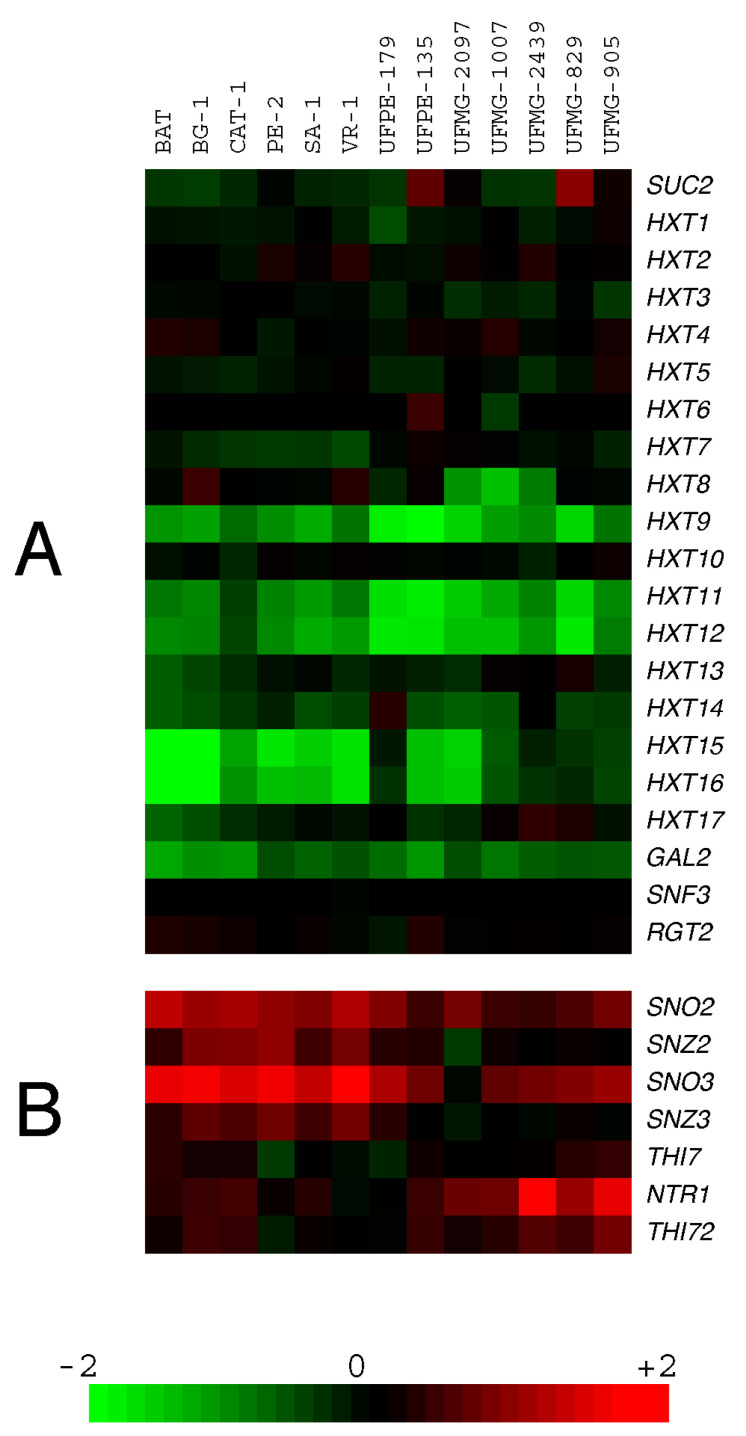
Copy number differences and similarities among industrial sugarcane yeast strains for genes involved in the metabolism and transport of sugar, and of vitamins B1 and B6: (**A**) array-CGH data showing the copy number of genes involved in sugar fermentation among the different strains; (**B**) array-CGH data showing the copy number of genes involved in thiamine (vitamin B1) and pyridoxine (vitamin B6) biosynthesis and transport among the different strains. A scale of relative gene copy number is shown at the bottom.

**Figure 2 jof-09-00803-f002:**
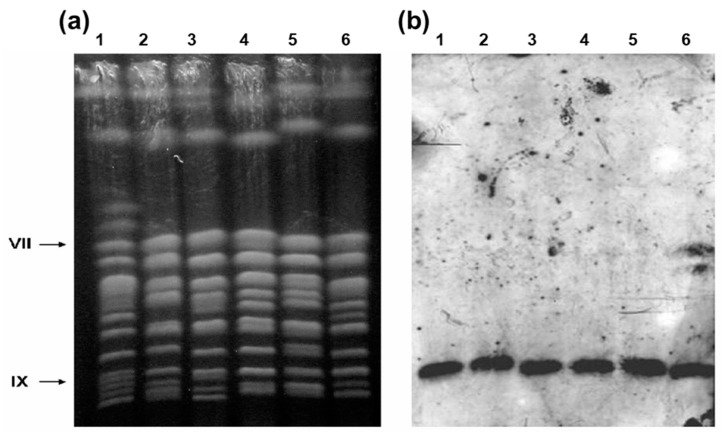
Detection of *SUC* genes in the industrial sugarcane yeasts. (**a**) PFGE separation of yeast chromosomes (ethidium-bromide stained). (**b**) Southern blot of gel shown in panel (**a**), hybridized with a probe for *SUC2* to detect which chromosomes carry *SUC* genes. (Lane 1) Reference laboratory strain S288C, which contains *SUC2* on chromosome IX (indicated to the left of panel (**a**)); (lanes 2–6) strains CAT-1, PE-2, BG-1, SA-1, and UFPE-136, respectively. This last strain has both the *SUC2* gene in chromosome IX and the *SUC1* gene in chromosome VII.

**Figure 3 jof-09-00803-f003:**
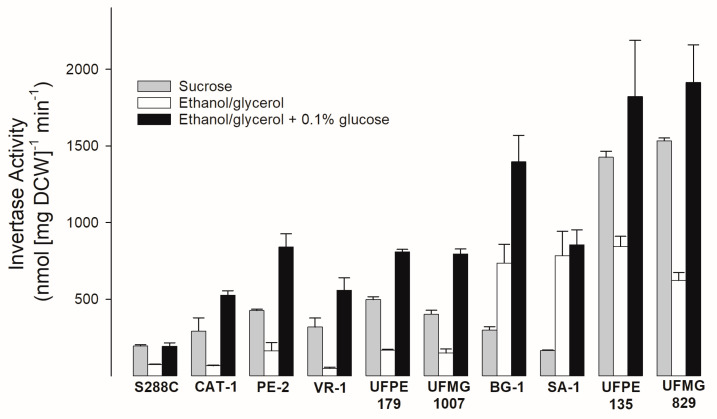
Invertase activity of the sugarcane industrial yeast strains. The extracellular invertase activity was determined after the growth of the cells in rich medium containing 20 g/L sucrose, 20 g/L ethanol plus 30 g/L glycerol (derepressed conditions), or in these derepressed conditions with media supplemented with 1 g/L glucose.

**Figure 4 jof-09-00803-f004:**
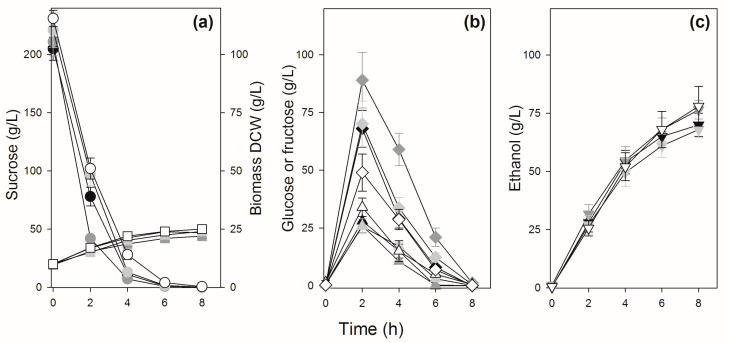
Sucrose batch fermentation by selected sugarcane industrial yeast strains. Panel (**a**) shows the concentrations of sucrose (circles) and biomass (squares), panel (**b**) shows the concentrations of glucose (triangles) and fructose (diamonds), while panel (**c**) shows the concentrations of ethanol (inverted triangles) during the fermentation by strain PE-2 (white symbols), UFPE-179 (light gray symbols), BG-1 (gray symbols), and UFMG-1007 (black symbols).

**Figure 5 jof-09-00803-f005:**
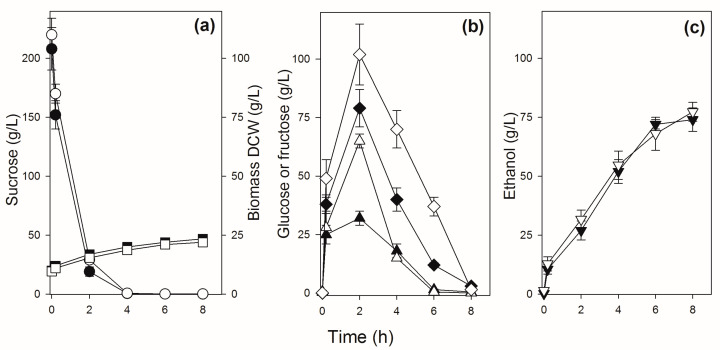
Sucrose batch fermentation by sugarcane industrial yeast strains with amplification of the *SUC* genes. Panel (**a**) shows the concentrations of sucrose (circles) and biomass (squares), panel (**b**) shows the concentrations of glucose (triangles) and fructose (diamonds), while panel (**c**) shows the concentrations of ethanol (inverted triangles) during the fermentation by strain UFMG-829 (white symbols) and UFPE-135 (black symbols).

**Figure 6 jof-09-00803-f006:**
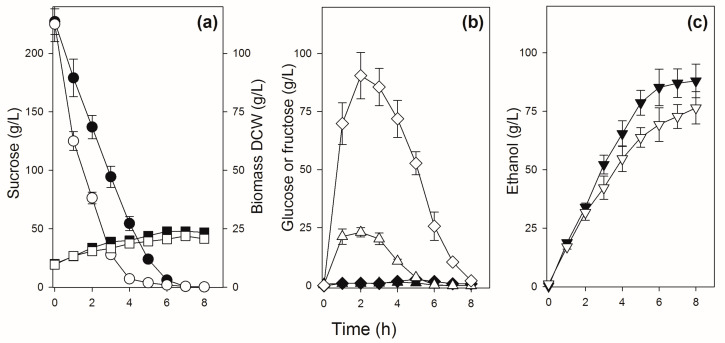
Sucrose batch fermentation by strains CAT-1 and GMY08. Panel (**a**) shows the concentrations of sucrose (circles) and biomass (squares), panel (**b**) shows the concentrations of glucose (triangles) and fructose (diamonds), while panel (**c**) shows the concentrations of ethanol (inverted triangles) during the fermentation by strain CAT-1 (white symbols) or GMY08 (black symbols).

**Table 1 jof-09-00803-t001:** Yeast strains used in this study.

Yeast Strains	Relevant Features	Source
BAT	Industrial fuel-ethanol strain isolated in 2011 from *Usina Batatais*, São Paulo, Brazil.	[40]
BG-1	Industrial fuel-ethanol strain isolated in 1989/1990 from *Usina Barra Grande*, Sao Paulo, Brazil.	[41]
CAT-1	Industrial fuel-ethanol strain isolated in 1998/1999 from *Usina VO Catanduva*, São Paulo, Brazil.	[12,41]
PE-2	Industrial fuel-ethanol strain isolated in 1993/1994 from *Usina da Pedra*, Sao Paulo, Brazil.	[12,41]
SA-1	Industrial fuel-ethanol strain isolated in 1989/1990 from *Usina Santa Adelia*, Sao Paulo, Brazil.	[41]
S288C	*MAT*α *mal gal2 mel flo1 flo8-1 hap1 SUC2*	[40]
BSY21-34B3	*MATa ura3-52 trp-289 kanMX*-P_ADH1_::*iSUC2*	[32]
UFMG-829	*Cachaça* strain isolated in 1996 from a distillery in Porto Firme, Minas Gerais, Brazil.	[13]
UFMG-905	*Cachaça* strain isolated in 1996 from a distillery in Nova União, Minas Gerais, Brazil.	[13,42]
UFMG-1007	*Cachaça* strain isolated in 1996 from a distillery in Salinas, Minas Gerais, Brazil.	[13]
UFMG-2097	*Cachaça* strain isolated in 1999 from a distillery in Salinas, Minas Gerais, Brazil.	[43]
UFMG-2439	*Cachaça* strain isolated in 1999 from a distillery in Salinas, Minas Gerais, Brazil.	[43]
UFPE-135	Industrial fuel-ethanol strain isolated in 1998/1999 from *Japungu* distillery, Paraíba, Brazil	[10]
UFPE-179	Industrial fuel-ethanol strain isolated in 1998/1999 from *Miriri* distillery, Paraíba, Brazil	[10]
VR-1	Industrial fuel-ethanol strain isolated in 1993/1994 from *Usina Vale do Rosario*, Sao Paulo, Brazil	[12,41]
GMY08	Isogenic to CAT-1, but *kanMX*-P_ADH1_::*iSUC2/suc2*Δ::*Ble*^R^ P_TDH3_::*AGT1*/*AGT1*	This work

**Table 2 jof-09-00803-t002:** Plasmids and primers used in this study.

	Relevant Features or Sequence (5′ → 3′)
Plasmids:	
pGRSd-AGT1	*ori amp^r^ CEN6 URA3* P_GPD_*-AGT1-*T_PGK_ [47]
pUG66	*ori amp^r^ LoxP*-*Ble*^R^-*LoxP* [48]
Primers: ^1^	
SUC100-F	GCGATAGACCTTTGGTCCAC
SUC1320-R	GGACCGTGGTAACTCTAAGG
V-SUC2F	GAAATTATCCGGGGGCGAAG
V2-SUC2F	GAGTTGTTGTCCTAGCGTAG
V2-SUC2R	TCCATTTCCCTCACTACTTC
V3-SUC2F	GCATCCACACGTCACAATCT
SUC100-R	GTGGACCAAAGGTCTATCGC
ssSUC2-F1	ATGCTTTTGCAAGCTTTCCTTTTCCTTTTGGCTGGTTTTGCAGCCAAAATATCTGCATCAGCCAGCTGAAGCTTCGTACGC
551SUC2-R1	ATTCGTATTGGTAGCCTAAGAAACCTTCATTGGCAAATGCAGATTCTAGCTTCCAGGACTGCATAGGCCACTAGTGGATC
GPD-AGT1F	GCCATAGATTCTACTCGGTCTATCTATCATGTAACACTCCGTTGATGCGTACTAGA**GAGTTTATCATTATCAATAC**
AGT1-389R	GAAAAACTGGCAGGGCATAC
V-GPDF	CAACCATCAGTTCATAGGTC
AGT1-I505-R	ACGGGCCAGCACTATAGTCTTAGTTCTC
V-AGT1F1	GAATTTTCGGTTGGTG

Underlined sequences allow amplification of the transformation module present in plasmid pUG66 [48], bold sequence allows amplification of the P_GPD_ promoter present in plasmid pGRSd-AGT1 [47].

**Table 3 jof-09-00803-t003:** Invertase activity, activity of the *AGT1* permease, and glycerol produced by the indicated yeast strains.

Strain	Activity ^1^ (nmol of Product [mg DCW]^−1^ min^−1^)			Glycerol Produced at the End of Fermentation ^2^ (g/L)
	Extracellular Invertase	Intracellular Invertase	*p*NPαG Transport	
CAT-1	295 ± 70	74 ± 6	0.1 ± 0.1	10.5 ± 1.2
GMY08	56 ± 2	2223 ± 45	4.5 ± 0.3	5.4 ± 0.7

^1^ Determined with cells grown with 20 g/L sucrose. ^2^ Fermentation of >200 g/L sucrose.

## Data Availability

All data is contained within the article and Appendix A.

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
