# Peer review of "Improved Sugarcane-Based Fermentation Processes by an Industrial Fuel-Ethanol Yeast Strain"

_jof, 2023, doi:10.3390/jof9080803_

Round 1

Reviewer 1 Report

This study analyzed the phenotypic and genetic characteristics of a group of eight bioethanol and five cachaça industrial yeasts to develop a strategy to genetically modify an industrial bioethanol strain for improved ethanol yield from sucrose. The modified industrial yeast strain consumed sucrose directly, without releasing glucose or fructose into the medium, and produced more ethanol when compared to the parental unmodified yeast strain. The below comments need to be considered by the authors:

1.       The title is too lengthy, please shorten it

2.       In the abstract, the problem statement should be written before the aim like why we need this study. what is the reason behind this study? So, one lined problem statement should be added at the start of the abstract.

3.       The originality and novelty of the paper need to be further clarified. The present form does not have sufficient arguments to justify the novelty of the paper.

4.       The literature review section should be improved. It should be dedicated to presenting a critical analysis of state-of-the-art related work to justify the objective of the study. Also, critical comments should be made on the results of the cited works.

5.       It is recommended to extend the comparison of the study findings with other similar published work under the results and discussion section. Currently, such a comparison is limited in this section. Moreover, add another or more sub-sections. It will make the discussion healthier and more attractive for the readers.

6.       It is strongly recommended to add a subsection, 'practical implications of this study,' outlining the challenges in the current research, future work, and recommendations, before the conclusion.

7.       Improve the research highlights and make them more results and objective oriented. Currently, all the highlights are about the study's aims/objectives.

8.       Proofreading should be conducted to improve both language and organization quality.

9.       It is recommended to provide a nomenclature or abbreviation table/box at the start of the manuscript, explaining all the abbreviated words used throughout the manuscript.

It's ok

Author Response

This study analyzed the phenotypic and genetic characteristics of a group of eight bioethanol and five cachaça industrial yeasts to develop a strategy to genetically modify an industrial bioethanol strain for improved ethanol yield from sucrose. The modified industrial yeast strain consumed sucrose directly, without releasing glucose or fructose into the medium, and produced more ethanol when compared to the parental unmodified yeast strain. The below comments need to be considered by the authors:

We thank the reviewer for his comments/suggestions, which certainly improved the quality of the manuscript. Below is a point to point answer to all comments.

  1. The title is too lengthy, please shorten it

Answer: as also suggested by the other reviewer, the little was shorted.

  1. In the abstract, the problem statement should be written before the aim like why we need this study. what is the reason behind this study? So, one lined problem statement should be added at the start of the abstract.

A: The abstract has been modified, hoping that the aims of the study are more clear.

  1. The originality and novelty of the paper need to be further clarified. The present form does not have sufficient arguments to justify the novelty of the paper.

A: In several points of the manuscript we have compared what was done in the past (use of laboratory vcs industrial strains, the need of laboratory evolution to obtain the improved strains, etc), while with 3 direct modifications we obtained a strain with a efficient and different mode of sucrose fermentation.

  1. The literature review section should be improved. It should be dedicated to presenting a critical analysis of state-of-the-art related work to justify the objective of the study. Also, critical comments should be made on the results of the cited works.

A: We have included several related work (with comments) in the Introduction (see lines 99-114)

  1. It is recommended to extend the comparison of the study findings with other similar published work under the results and discussion section. Currently, such a comparison is limited in this section. Moreover, add another or more sub-sections. It will make the discussion healthier and more attractive for the readers.

A: As suggested also by the other reviewer, we have now combined the results with discussion, and added a new section.

  1. t is strongly recommended to add a subsection, 'practical implications of this study,' outlining the challenges in the current research, future work, and recommendations, before the conclusion.

A: This new section was included (lines 486-499)

  1. Improve the research highlights and make them more results and objective oriented. Currently, all the highlights are about the study's aims/objectives.

A: It was not clear for us were the “research highlights” are? in many parts we have highlight the results obtained.

  1. Proofreading should be conducted to improve both language and organization quality.

A: Done

  1. It is recommended to provide a nomenclature or abbreviation table/box at the start of the manuscript, explaining all the abbreviated words used throughout the manuscript.

A: We did not found any instructions on how to prepare/format such table/box with abbreviations. All few abbreviations used (“YP”, “PFGE” or “array-CGH”) were described upon their first appearance in the text, other “abbreviations” are gene names!

Reviewer 2 Report

-Remove this from your title "Comparative Genomic Analysis and Phenotypic Characterization of Saccharomyces cerevisiae Strains used in"

-Rephrase as: "Improved Sugar-cane-based Fermentation Processes by an Industrial Fuel-ethanol Yeast Strain"

-Re-write the abstract to include statistical results and let there be a flow of the sentences.

-All these are not keywords.

Suggested keywords: 

-Sugarcane

-fermentation

-Yeast

-Bioethanol

--The authors don't have to personalize by saying "we" repeatedly

-line 100: -Is the study a report or research?

-line 105-112: -The aim and objectives of the study is not clear

line 117: -did you measure 20g/L of this too? Then say it

section 2.1: HCl and NaOH of what concentration?

line 164: Why? Or this....be specific

-Redraw figure4 and split the fig 4a, 4b and c. Let it look good and with reasonable units

-Redraw figure 5 and split the fig 5a, 5b and 5c. Let it look good and with reasonable units.

-If the journal allows you to combine result and discussion please do?

line 479-483: Future studies or upscaling that talked about optimization studies should be recommended and included in your study.

Author Response

We thank the reviewer for his comments/suggestions, which certainly improved the quality of the manuscript. Below is a point to point answer to all comments.

-Remove this from your title "Comparative Genomic Analysis and Phenotypic Characterization of Saccharomyces cerevisiae Strains used in" Rephrase as: "Improved Sugar-cane-based Fermentation Processes by an Industrial Fuel-ethanol Yeast Strain"

Answer: The title was changed as suggested

-Re-write the abstract to include statistical results and let there be a flow of the sentences.

A: several parts of the abstract were changed

-All these are not keywords. Suggested keywords: Sugarcane, fermentation. Yeast, Bioethanol

A: keyword changed as suggested.

--The authors don't have to personalize by saying "we" repeatedly

A: all “we” were removed from the manuscript.

-line 100: -Is the study a report or research?

A: The sentence was changed (now line 115)

-line 105-112: -The aim and objectives of the study is not clear

A: We have change this part, hoping that the objectives are more clear (lines 106-128)

-line 117: -did you measure 20 g/L of this too? Then say it

A: included as requested (now line 133)

-section 2.1: HCl and NaOH of what concentration?

A: we included the concentration of the solutions: 1 M for HCl and 0.5 M for NaOH (lines 137-138)

-line 164: Why? Or this....be specific

A: the sentence was changed and more specific (now lines 184-185)

-Redraw figure 4 and split the fig 4a, 4b and c. Let it look good and with reasonable units

-Redraw figure 5 and split the fig 5a, 5b and 5c. Let it look good and with reasonable units.

A: Figures 4, 5 and 6 were modified as requested.

-If the journal allows you to combine result and discussion please do?

A: We have now combined the results with discussion

-line 479-483: Future studies or upscaling that talked about optimization studies should be recommended and included in your study.

A: a final section was included discussing future studies, etc (lines 486-499)